# Bt Trait Efficacy Against Corn Earworm, *Helicoverpa zea*, (Lepidoptera: Noctuidae) for Preserving Grain Yield and Reducing Mycotoxin Contamination of Field Corn

**DOI:** 10.3390/insects15120914

**Published:** 2024-11-22

**Authors:** William Yancey Barton, George David Buntin, Micheal D. Toews

**Affiliations:** 1Georgia Department of Agriculture, Atlanta, GA 30334, USA; yancey.barton@agr.georgia.gov; 2Department of Entomology, University of Georgia-Griffin Campus, Griffin, GA 30223, USA; 3Department of Entomology, University of Georgia-Tifton Campus, Tifton, GA 31793, USA; mtoews@uga.edu

**Keywords:** *Zea mays*, transgenic crop, maize, *Bacillus thuringiensis*, fumonisin, aflatoxin, Vip3Aa20

## Abstract

The corn earworm causes persistent ear damage to corn grown in the southeastern United States. Increased levels of ear damage have been associated with mycotoxin contamination such as aflatoxin and fumonisin. Corn hybrids expressing the Bt traits Vip3Aa20 provided substantial corn earworm control and prevented kernel damage. Older Bt corn without this trait did not control corn earworm in the ear and prevent ear damage. Bt corn that prevented kernel damage had a variable effect on grain yield but may prevent yield loss. Bt corn that prevented ear damage did not suffer grain contamination from aflatoxin but did show reduced grain contamination by fumonisin.

## 1. Introduction

Corn, *Zea mays* L., genetically modified with proteins from the *Bacillus thuringiensis* (Bt) bacterium, was commercially released in North America as a method of controlling stalk-boring pests including the European corn borer, *Ostrinia nubilalis* (Hübner) (Lepidoptera: Crambidae) [1,2]. This method ultimately provided mixed results when tested against corn earworm, *Helicoverpa zea* (Boddie), a lepidopteran pest species with a wide agronomic host range including field corn. Corn earworm larvae feed on the corn silks, ears and kernels for six instars prior to pupating in the soil, resulting in significant damage to the upper ear region. However, the impact of larval feeding alone is generally not a significant source of yield loss in field corn production [3,4,5] since most corn earworm activity occurs at the ear tip region where unpollinated kernels are located [6]. However, significant ear damage with some level of grain yield reduction has been reported in late plantings when larval activity is greater [7,8,9,10].

Bt traits produce plant proteins and are widely incorporated into transgenic corn hybrids throughout the United States, with 82% of all domestically planted corn in 2020 having expressed single or pyramided Bt toxins [11]. Early studies in the southeastern United States region showed that Cry1Ab in events BT11 and MON810 provided partial control against corn earworm infestations, but significant control and improved yield was mostly observed for whorl infestations by fall armyworm, *Spodoptera frugiperda* (J. E. Smith) (Lepidoptera: Noctuidae) [7,8,12,13,14]. Additional Bt toxins became commercially available as pyramids with varying levels of efficacy for corn earworm control. The MON89034 event expresses the pyramided proteins Cry1A.105 + Cry2Ab2 and when released provided better control of corn earworm than Cry1Ab [3,8,14,15,16]. However, resistance to these toxins developed across local lepidopteran pest populations. For corn earworm, resistance to Cry1Ab is widespread and resistance to Cry2Ab was reported at local levels across several southern states [17,18,19,20]. The MIR162 event in corn expressing the Vip3Aa20 insecticidal protein currently is the most effective transgenic event for reducing corn earworm numbers in corn [9,18,21,22,23]. Indeed, currently corn expressing the vip3Aa20 protein almost completely eliminates ear infestation by corn earworm [9,20,24]. Nevertheless, the presence of resistant alleles in corn earworm to Vip3Aa20 protein has been confirmed in localized populations from Texas and the Mid-South. There is evidence that resistance to Vip3Aa20 may be increasing in field-collected corn earworms [24,25].

Lepidopteran damage to corn ears may result in corn becoming vulnerable to toxic secondary metabolites known as mycotoxins [26,27,28,29]. The fungus *Aspergillus flavus* (Link) (Deutermycetes: Moniliales) will express the mycotoxin aflatoxin B_1_ [29,30,31]. Corn grain in semi-tropical areas also may be infected by *Fusarium* spp. with *Fusarium verticillioides* (Sacc.) Nirenberg (Sordariomycetes: Hypocreales) being the primary species expressing the mycotoxin fumonisin [28,32]. Grain that is highly contaminated with aflatoxin or fumonisin may be toxic when consumed by humans and animals. A major mycotoxicosis outbreak that occurred across rural Kenya in 2004 was the result of post-harvested corn storage under damp conditions that promoted severe grain aflatoxin contamination, resulting in numerous cases of sickness and death [33]. In addition to excessive moisture during storage, heightened mycotoxin contamination can occur from other factors in the field including high temperatures, drought conditions, and plant genetics [29,34,35,36,37]. The U.S. Food and Drug Administration (FDA) limits in grain intended for human consumption are no more than 20 ppb for aflatoxin and 2–4 ppm for fumonisin. Grain intended for animal feed can have higher limits depending on the animal age, weight, and production stages [38,39]; for example, the FDA aflatoxin limit for breeding swine is 200 ppb. Early reports found that increased aflatoxin contamination levels were linked with significant ear injury caused by lepidopteran pests, including corn earworm [26,40]. Increased larval feeding could promote mycotoxin contamination through induced plant stress [30,31,34,37] or the larvae may simply harbor fungal spores within their gut that are transferred into host plants upon feeding [41]. Furthermore, planting date plays a significant role in fungal susceptibility for field corn production in temperate and tropical regions as later planted corn endures more stress [32,34,35,36].

Both aflatoxin and fumonisin contamination have been associated with ear feeding by larvae of the corn earworm, fall armyworm and several other lepidopterans [31,32,37]. The overall effect of transgenic Bt hybrids in reducing aflatoxin contamination is inconsistent. Multiple studies found significant associations between lepidopteran pest ear damage and aflatoxin contamination levels in transgenic corn [31,34,35,37,42,43]. However, other studies found greater inconsistent results or no clear association despite making similar comparisons [4,7,14,44,45,46]. Nevertheless, a study using aflatoxin-related insurance claims in the U.S. found that aflatoxin risk was lower in counties where more Bt corn was planted [47]. Although adoption of Bt hybrids may contribute to aflatoxin mitigation [37,47], it remains inconclusive how effective Bt technology is as a tool for aflatoxin reduction in field corn [27].

Conversely, fusarium ear rot and fumonisin contamination have a more definite association with increased ear injury from lepidopteran pest feeding. In Iowa, Munkvold et al. [48] found increased European corn borer injury induced *Fusarium* ear rot, and Bowers et al. [49,50] found positive associations in fumonisin accumulations with increased ear injury from European corn borer, corn earworm and western bean cutworm, *Striacosta albicosta* (Smith) (Lepidoptera: Noctuidae). Western bean cutworm is associated with deoxynivalenol and Gibberella ear rot from accumulated *Fusarium graminearium* infection of corn in the midwestern United States and Ontario, Canada [51,52]. Parson and Munkvold [32] also found a correlation between lepidopteran kernel damage and fumonisin B_1_ contamination in corn. A meta-analysis of studies on genetically engineered maize found that hybrids expressing Bt toxins exhibited lower concentrations of mycotoxins by 28.8% and fumonisins by 30.6% [53]. Incorporating Bt technology could potentially reduce grain fumonisin contamination [27,36,52,53], especially use of newer, more effective pyramided hybrids that express the Vip3Aa20 protein [48,52].

The objective was to evaluate how commercially available Bt hybrids expressing various Bt proteins prevented corn earworm ear damage and mycotoxin contamination. A selection of commercially available non-Bt hybrids as well as hybrids expressing various pyramided Bt traits was evaluated. Larval infestations and ear damage were measured and correlated with grain yield loss or grain mycotoxin contamination levels to determine whether Bt technologies preserved grain yield and quality in the field. The authors hypothesized that only those Bt hybrids expressing the Vip3Aa20 protein will effectively reduce corn earworm infestations. Reductions in aflatoxin and fumonisin contamination are expected only where there is significant reduction in corn earworm ear and kernel damage.

## 2. Materials and Methods

### 2.1. Field Experiments

Field experiments were conducted at two locations per year in central and southern Georgia in 2019 and 2020. Locations were the University of Georgia Bledsoe Research farm near Griffin (N 33.175964 W −84.409210), the Southwest Georgia Research and Education Center near Plains (N 32.046602 W −84.370610), and the University of Georgia Lang-Rigdon farm near Tifton (N 31.516910 W −83.548479). Soil was an Appling sandy loam in Griffin, a Greenville sandy loam in Plains, and Tifton sandy loam in Tifton. Weed control and fertility practices followed the Georgia Extension Service recommendations for each location. Conventional tillage was used at all locations with chisel plowing followed by disk harrowing. Before disking, 500 kg/ha of a 5–10–15 (N-P-K) granular fertilizer was applied and an additional 112 kg of nitrogen as ammonium nitrate was applied beside the rows and incorporated about 20 days after planting. For weed control, atrazine (Aatrex 4L, Syngenta Crop protection, Greensboro, NC, USA) and pendimethalin (Prowl 3.3EC, BASF, Research Triangle Park, NC, USA) were applied at planting at all locations except in Griffin, where atrazine with acetochlor (Warrant, Bayer CropScience LP, St. Louis, MO, USA) was applied. Plots were treated with a broadcast application of glyphosate (Roundup WeatherMax, Bayer CropScience, St. Louis, MO, USA) about 20 to 25 days after planting for post-emergence weed control. All corn seed was received from the seed companies and pretreated with two or three fungicides and either clothianidin at 0.5 mg per kernel (Poncho 250 or 500, Bayer CropScience, St. Louis, MO, USA) or thiamethoxam at 0.5 mg per kernel (Syngenta Crop Protection, Greensboro, NC, USA). No other insecticides were applied. Natural rainfall was supplemented weekly by irrigation of 6 cm of water as needed.

A selection of available hybrids with various pyramided Bt proteins for above-ground pests were evaluated and compared with non-Bt hybrids of similar relative maturity and agronomic traits (Table 1). Hybrids were provided by DeKalb Seeds (Bayer CropScience, St. Louis, MO, USA) and Pioneer Hi-bred International Inc. (Corteva AgriScience, Johnston, IA, USA). Planting dates in 2019 were 24 April in Plains and 25 April in Tifton, and in 2020 were 3 April in Tifton and 16 May in Griffin. Experimental design for all plantings was a randomized complete block design with four replicates. Corn seed was planted at a rate of 79,040 seeds per ha in 91 cm wide rows at the Plains and Tifton locations and 76 cm wide rows at the Griffin location using a two-row Monosem^®^ pneumatic planter (Largeasse, France). Plots were eight rows wide and 12.2 m long, except in the 2020 Tifton plantings, which were eight rows wide and 10.7 m long.

### 2.2. Data Collection

Plant stand counts were made on the center two rows per plot while inspecting for whorl damage on the corn plants caused by lepidopteran defoliators, primarily fall armyworm larvae, at the five- to eight-leaf vegetative growth stages. The percentage of damaged plants and severity of damage was rated using the Davis et al. [54] 0–9 scale, where 0 represents no damage and 9 represents nearly total destruction of the whorl. Selected infested plants in border rows were inspected to determine the identification of whorl-infesting larvae.

Evaluation for corn earworm infestations occurred when crops reached the milk stage (R3) [55]. Fifteen random ears from each plot were opened from their husks and examined for the presence of corn earworm larvae. The total number of larvae was counted and categorized as either small (first and second instars), medium (third and fourth instars), or large (fifth and sixth instars) in size. Small larvae were identified as no larger than 7 mm, medium larvae were 8 to 24 mm long, and large larvae were greater than 24 mm in length. Exit holes with underlying ear damage also were counted, indicating that a larva had completed development and exited the ear to pupate in the soil. Plots were evaluated for total corn earworm damage at the late dent stage (R5) to early physiological maturity stage (R6) [55]. Another 15 random ears were inspected in each plot and the total area damage from larval feeding was measured in cm^2^. Damage measurements for every ear were divided between the tip portion with unpollinated kernels and the rest of the ear containing viable kernels.

At full maturity, grain was harvested from the center two rows not used for earworm sampling of each plot using a two-row self-propelled Wintersteiger Delta combine (Wintersteiger Inc., Salt Lake City, UT, USA) with an automated weighing system that measured plot grain weight, percentage moisture content, and test weight. Grain yields were adjusted to a standard 15.5% moisture content and extrapolated to a kg/ha basis. A 1-kg sample grain was collected from each plot and processed by Waters Agricultural Laboratories, Inc. (Camilla, GA, USA) to measure grain mycotoxin contamination levels. The NEOGEN Veratox^®^ method, a competitive direct ELISA, provided quantitative analyses of total aflatoxin (B_1_, B_2_, G_1_, G_2_) (ppb) and total fumonisin (B_1_, B_2_, B_3_) (ppm) contamination levels of corn grain [56,57]. Both methods are approved by the AOAC Research Institute [58]. Samples that read higher levels than the standard provided in each kit were diluted then reanalyzed with the additional dilution factor applied to the quantitative value.

### 2.3. Statistical Analyses

The statistical software SAS version 9.4 and JMP Pro version 15.0.0 were used for data analysis [59,60]. Results were generalized across locations but were analyzed separately by year because some hybrids exhibited wide variability between 2019 and 2020. Results were statistically analyzed through an analysis of variance (ANOVA) mixed model with PROC MIXED and appropriate statistical procedures for a randomized completed block design including hybrids coded as a fixed effect and replicates coded as a random effect with the Kenward–Roger degrees of freedom approximation option. Before analysis, percentage values were transformed with an angular transformation before analysis and larva counts and damage and mycotoxin values were transformed with a log10(x + 1) transformation to normalize variances. Non-transformed values are presented in the results and figures. Single degree-of-freedom contrasts were used to compare the responses of hybrids expressing Bt proteins with non-Bt hybrids, Bt hybrids expressing only Cry proteins with non-Bt hybrids, Bt hybrids expressing Vip3Aa20 with non-Bt hybrids, and Bt hybrids expressing only Cry proteins with hybrids expressing Cry + Vip3Aa20 proteins. When hybrid treatment *F* tests were significant, means were separated using pairwise *t*-test groupings in PROC PLM when a significant difference was indicated among *F*-values (α = 0.05). Linear regression of a bivariate fit of two continuous data type variables using PROC REG [59] was used to test associations of corn earworm kernel damage with grain yield and contamination levels of total grain aflatoxin and fumonisin.

## 3. Results

### 3.1. Infestation Rates and Ear Damage

Little whorl defoliation was observed during the vegetative growth stages in all experiments and the small number of non-Bt plants with whorl damage was caused by fall armyworm infestation. Overall corn earworm infestations and damage ratings were lower than normal for field corn in southern Georgia for both 2019 and 2020. Infestation levels were significantly higher in those experiments planted from late April into May than those planted in early April. Corn earworm infestations were significantly different among hybrids in 2019 (*F* = 96.98, df = 9, 54; *p* < 0.0001) and 2020 (*F* = 25.85, df = 11, 66; *p* < 0.0001) (Table 2 and Table 3). Overall, Bt hybrids reduced corn earworm infestations during the R3 growth stage by an average of 65.2% in 2019 (non-Bt: 73.12 ± 3.53% infested plants; Bt: 25.42 ± 4.53% infested plants) and 53.6% in 2020 (non-Bt: 53.33 ± 5.08% infested plants; Bt: 24.76 ± 4.35% infested plants). The Bt products Genuity Trecepta and Optimum Leptra containing Cry genes and the Vip3Aa20 gene had the lowest levels of larval infestation in every experiment with very little to no infestation at R3 growth stage in both years (Table 2 and Table 3). Bt hybrids expressing only Cry proteins provided intermediate infestation control, but infestation levels varied in statistical significance from non-Bt hybrids. Genuity VT Double PRO (Cry1A.105 + Cry2Ab2) and SmartStax (Cry1A.105 + Cry2Ab2 + Cry1Fa2) reduced corn earworm infestations in 2019, but only SmartStax had reduced infestation in 2020. Infestations of Optimum Intrasect (Cry1Ab + Cry1Fa2) were not significantly different from those in comparable non-Bt hybrids.

The results for average corn earworm larvae per ear were similar to those of infestation levels. Hybrids exhibited significantly different numbers of total larvae per ear among hybrids in 2019 (*F* = 48.82; df = 9, 54; *p* < 0.0001) and 2020 (*F* = 17.39; df = 11, 66; *p* < 0.0001). The Pioneer brand hybrids as a group tended to have more larvae per ear than the Dekalb hybrids (Figure 1 and Figure 2). A total of only five larvae and three exit holes were observed in plants of all hybrids expressing Vip3Aa20 in all experiments with almost all of these being in the early planting in 2020. Bt hybrids expressing only Cry proteins with the Cry2Ab2 protein had significantly fewer larvae than non-Bt hybrids in both years, whereas total larvae counts were similar for the hybrids expressing Cry1Ab + Cry1Fa2 and comparable non-Bt hybrids in both years (Figure 1 and Figure 2). The number of larvae within each size category was significantly different for all four size categories in 2019 (*F* = 12.41 to 15.72; df = 9, 54; *p* < 0.0001) and 2020 (*F* = 4.11 to 6.18; df = 11, 66; *p* < 0.001). Larvae from hybrids expressing only Cry proteins were mostly small or medium in size as compared with larvae in ears of non-Bt hybrids that were mostly large or had already exited the ear (Figure 1 and Figure 2; Appendix A).

Results for R6 growth stage ear damage were also significantly different between hybrids (Table 2 and Table 3). The proportion of damage in the tip and kernel regions was similar in 2019, but in 2020 much more kernel damage occurred relative to tip ear damage. Overall Bt hybrids reduced total corn earworm ear damage by an average of 75.0% in 2019 (non-Bt: 4.16 ± 0.32 cm^2^; Bt: 1.04 ± 0.26 cm^2^) and 66.7% in 2020 (non-Bt: 3.18 ± 0.30 cm^2^; Bt: 1.06 ± 0.25 cm^2^). Bt hybrids expressing Vip3Aa20 were undamaged in 2019 and exhibited only minimal ear damage, located almost entirely within the ear tip region, during the early planting in 2020 (Table 2 and Table 3). Bt hybrids expressing only Cry proteins with the Cry2Ab2 protein provided moderate but significant reductions in both tip and kernel damage compared to non-Bt hybrids but were not as effective as hybrids expressing Vip3Aa20 (Table 2 and Table 3). Hybrids with Cry1Ab + Cry1F reduced tip and kernel damage in 2019 but did not reduce damage in 2020.

### 3.2. Grain Yield and Test Weight

Grain yields differed among hybrids in 2019, but these differences were not associated with Bt traits (*F* = 0.02; df = 1, 54; *p* = 0.8764) and instead reflected hybrid agronomics (Table 4). In 2020, grain yields were significantly different among hybrids and all Bt hybrids produced an average of 5.84% more grain yield than non-Bt hybrids (non-Bt: 12,642 ± 2270 kg/ha; Bt: 13,380 ± 1934 kg/ha; *F* = 11.95; df = 1, 66; *p* < 0.0009) (Table 5). Hybrids with Cry proteins and those with Vip3Aa20 both yielded more than the non-Bt hybrids (*p* = 0.0329 and *p* = 0.0044, respectively) but were not significantly different from each other (*F* = 2.82; df = 1, 66; *p* = 0.0975) (Table 5). Based on linear regression analysis, grain yield was not associated with corn earworm ear damage in 2019 (R^2^ = 0.0061, *F* = 0.4791, *p* = 0.4909) but had a significantly negative association with corn earworm damage in 2020 (R^2^ = 0.2638, *F* = 33.690, *p* < 0.0001) (Appendix A).

### 3.3. Mycotoxin Contamination Levels

Grain aflatoxin contamination in 2019 and 2020 was highly variable among plot samples. While most concentrations remained around the federal standard limit of aflatoxin in grain intended for human consumption of 20 ppb, a small number of samples exceeded 100 ppb with a few exceeding 500 ppb. Contamination levels were significantly different among hybrids, most likely due to genetic background, because aflatoxin levels were not different between hybrids with and without Bt traits in 2019 (*F* = 0.57; df = 1, 54; *p* = 0.4524) and 2020 (*F* = 0.03; df = 1, 66; *p* = 0.8543) (Table 4 and Table 5). Variability in concentrations for a select few points resulted in skewing of statistical significance among hybrids. The Bt hybrid Pioneer 1637YHR, an Optimum Intrasect product, had an average aflatoxin contamination level significantly greater than those of all other hybrids in 2019 (Table 4). Linear regression analyses showed that aflatoxin contamination levels were not significantly associated with corn earworm damage in either year (2019: R^2^ = 0.0158, *F* = 1.2517, *p* = 0.2667; 2020: R^2^ = 0.0249, *F* = 2.4050, *p* = 0.1243) (Figure 3).

Grain fumonisin contamination in 2019 was variable among plots but was significantly different among hybrids (Table 4). Only Bt hybrids expressing the Vip3Aa20 protein had significantly lower contamination levels compared to non-Bt hybrids (*F* = 17.49, df = 1, 54; *p* < 0.0001). However, grain fumonisin contamination levels for all hybrids in 2019 exceeded the federal standard limit of 2–4 ppm for grain intended for human consumption [39]. Bt hybrids expressing only Cry proteins did not have significantly reduced fumonisin contamination as compared to non-Bt hybrids (*F* = 1.11, df = 1, 54; *p* = 0.2962), and were not statistically different from Bt hybrids expressing Vip3Aa20 (*F* = 1.93, df = 1, 54; *p* = 0.1707). Overall contamination levels in 2020 were much lower than in 2019 and were not significantly different among hybrids (Table 5). Grain fumonisin contamination levels in 2020 also were not significantly different among Bt traits (*F* = 0.02, df = 1, 66; *p* = 0.0900). Grain fumonisin contamination levels had a significant positive association with corn earworm ear damage in both 2019 and 2020, with the association being stronger in 2019 when contamination levels were higher (Figure 4).

## 4. Discussion

Planting transgenic corn hybrids that express multiple Bt traits through gene pyramiding has been a continuing protocol enacted in temperate and subtropical regions where lepidopteran pest infestation, including corn earworm, is common [14,16,49,61]. Resistance to specific Bt traits has been increasing in local corn earworm populations across the southeastern United States region, making them more difficult to control [17,19,20,42,62,63,64,65].

Bt corn hybrids expressing the Vip3Aa20 protein were the most effective in reducing corn earworm infestations and ear damage in all experiments. These results are consistent with other studies that show Bt products expressing Vip3Aa20 having minimal or no earworm infestations [19,21,64,65]. Bt hybrids expressing only Cry proteins provided moderate reductions in earworm infestation and damage. Despite having similar numbers of total corn earworm larvae in ears to those of non-Bt hybrids, Bt hybrids expressing only Cry proteins had mostly small and medium-sized larvae, while larvae from non-Bt hybrids were mostly large or were exiting the ears to pupate in the soil. This indicates that while the Cry protein did not cause much direct mortality, the toxins did delay the development of corn earworm larvae. It is unknown if any surviving corn earworms from these Bt hybrid plots had reduced adult survival and fecundity [66]. Bt hybrids with pyramided Cry proteins alone also caused partial reductions in both tip and kernel damage. Bt hybrids that expressed Cry1Ab + Cry1Fa2 proteins failed to prevent any ear infestation or damage caused by corn earworm larvae. Cry1Fa2 has not been considered an effective insecticidal source against corn earworm in the ear [67], and the potency of Cry1Ab has declined due to widespread resistance against the toxin in corn earworm populations across the southeastern United States region [18,19,20,63,64,66]. Partial earworm control at optimally timed plantings was most notable for Bt products expressing Cry1A.105 + Cry2Ab2, yet these products remain vulnerable to further resistance development [20,66,68]. These data suggest that Bt hybrids will continue to provide effective control of lepidopteran pest species in the southeastern U.S. including the European corn borer, fall armyworm, and corn earworm [7,8,48,66,67,69,70]. Future use of Cry proteins in regions with notable corn earworm activity must be carefully considered to extend and manage the efficacy of available Bt products.

While corn earworms are well-known for infesting ears and feeding on the kernels, the resulting injury normally does not qualify them as an economically significant pest of field corn [4,67,71]. Simulated corn earworm damage only resulted in significant yield loss when at least 60 kernels, or 15 cm^2^ based on the conversion 0.25 cm^2^ = 1 kernel, were damaged per ear [5]. Mean kernel damage in both 2019 and 2020 was much lower than the equivalent of 60 kernels per ear even in the late plantings. However, all Bt hybrids preserved significantly more grain yield than non-Bt hybrids in 2020, when ear damage in non-Bt hybrids was greatest and earworm damage was correlated with grain yield. While corn earworm currently may not be a primary issue in Georgia corn production, Bt hybrids can be effective in reducing kernel loss because of larval ear feeding [3,70] and may in some cases prevent grain yield loss, especially in later plantings [68]. Determining whether Bt hybrids could also improve grain quality based on test weight requires further evaluation.

Aflatoxin contamination accumulates from *Aspergillus* spp. infection and is notorious for emerging in an array of crops, including corn, whether transferred from pests [41] or developing through induced plant stress conditions including insect feeding injury [29,31,34,41,43]. Early studies suggest higher grain aflatoxin contamination levels occur in plants that have greater amounts of lepidopteran pest ear damage, especially by corn earworm [26,40]. Our data suggest that grain aflatoxin contamination was not associated with corn earworm ear damage nor was it reduced by the presence of any Bt toxins. Varying levels of fungal contamination were observed in similar studies that found a lack of association between corn earworm ear injury and grain aflatoxin levels [4,7,46,50]. Relative differences in aflatoxin contamination levels may instead be influenced by environmental factors such as high temperature and humidity, kernel moisture content, and rainfall across planting locations and dates [30,34,35]. Production systems that minimize plant stress conditions such as early planting and irrigation may assist with reducing grain aflatoxin contamination in subtropical regions.

Bt corn hybrids in this study effectively reduced corn earworm infestations and grain fumonisin contamination accumulated from *Fusarium* spp. infection based on linear regression analysis in both years. Past reports have found associations between fumonisin contamination with lepidopteran pest species including the European corn borer and western bean cutworm [27,48,51,52,72,73]. In a meta-analysis of 21 years of studies, Bt technology reduced fumonisin contamination in corn grain by 30.6% but this analysis included results for studies worldwide with a variety of target lepidopteran pest species [53]. Work specifically on the relationship between corn earworm damage and fumonisin contamination is limited. Bowers et al. [49] observed strong associations between increased fumonisin contamination and kernel injury resulting from three lepidopteran pest species that were all significantly reduced in corn expressing Cry1Ab + Vip3Aa20. Other studies also found an association with lepidopteran ear damage and fumonisin levels in subtropical locations, but they did not identify a specific lepidopteran species [32,36]. Here the authors provide supporting data on fumonisin contamination levels in corn hybrids expressing a wider selection of pyramided Bt proteins intended for lepidopteran pest control, which showed that Bt hybrids expressing Vip3Aa20 were associated with reduced grain fumonisin contamination levels. Nevertheless, overall contamination was still over the federal standard for grain intended for direct human consumption and only partially met the standard for certain animal feeds [39].

## 5. Conclusions

This study demonstrates the importance of Bt corn hybrids expressing the Vip3Aa20 protein for corn earworm control in the southeastern United States region. Planting transgenic Bt corn hybrids expressing Vip3Aa20 reduced grain fumonisin contamination but did not reduce aflatoxin contamination. Older Cry protein Bt products provided only limited control of corn earworm damage and did not consistently reduce mycotoxin contamination. Increased use of the Vip3Aa20 protein across more than one crop for controlling Cry-resistant *H. zea* populations could reduce its durability [74,75], thereby emphasizing the importance of insecticide resistance management tactics such as planting non-Bt refuges and careful management of pyramided Bt products to protect its durability [21,74,75,76,77,78]. Bt products containing Vip3Aa20 can be combined with improved corn genetics and agronomic practices that reduce crop stress to mitigate fumonisin contamination of corn grain.

## Figures and Tables

**Figure 1 insects-15-00914-f001:**
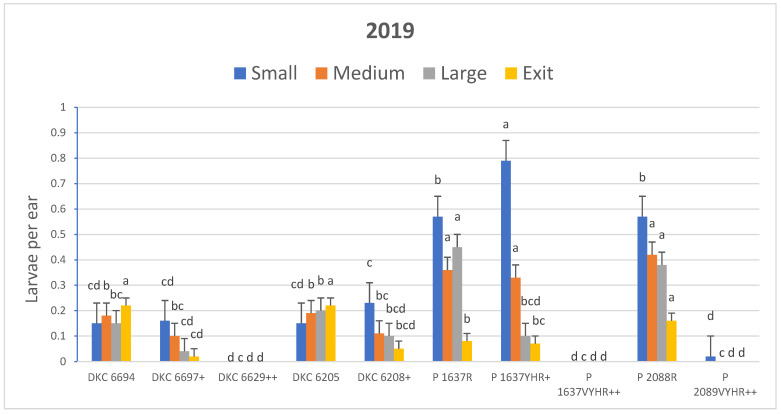
Effect of Bt traits on LS means ± SEM of number of corn earworm larvae per ear by size and exit holes in R3 growth stage 2019 field corn. LS means within larval size category with the same letter are not significantly different (pair-wise *t*-tests of LSM, α = 0.05). Bt hybrids marked with (+) expressed pyramided *Cry* proteins while hybrids marked with (++) expressed pyramided *Cry* proteins with Vip3Aa20. Unmarked hybrids were non-Bt hybrids. See Table 1 for specific proteins expressed for each entry and product type.

**Figure 2 insects-15-00914-f002:**
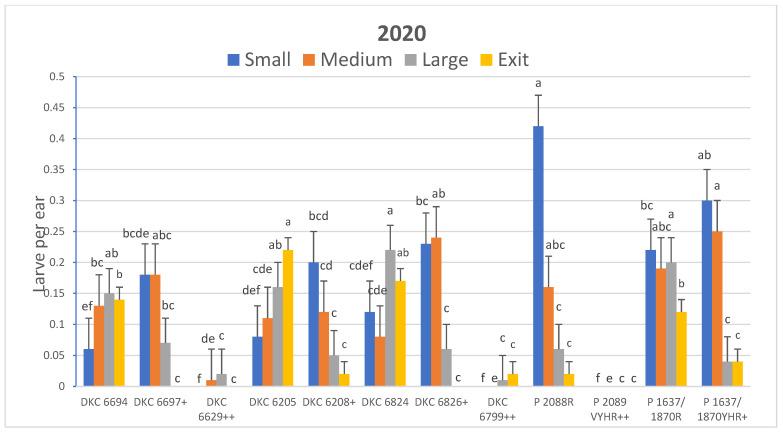
Effect of Bt traits on LS means ± SEM of number of corn earworm larvae per ear by size and exit holes in R3 growth stage 2020 field corn. LS means within larval size category with the same letter are not significantly different (pair-wise *t*-tests of LSM, α = 0.05). Bt hybrids marked with (+) expressed pyramided *Cry* proteins while hybrids marked with (++) expressed pyramided *Cry* proteins with Vip3Aa20. Unmarked hybrids were non-Bt hybrids. See Table 1 for specific proteins expressed for each entry and product type.

**Figure 3 insects-15-00914-f003:**
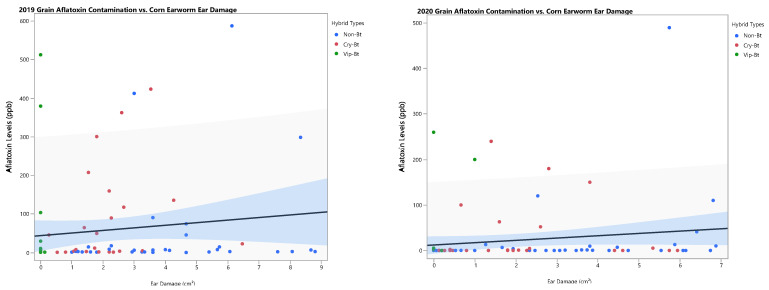
Linear regression analysis depicting the relationship between corn earworm ear damage and grain aflatoxin contamination levels of field corn by year. 2019: R^2^ = 0.0158; *F* = 1.2517; *p* = 0.2667. 2020: R^2^ = 0.0249; *F* = 2.4050; *p* = 0.1243.

**Figure 4 insects-15-00914-f004:**
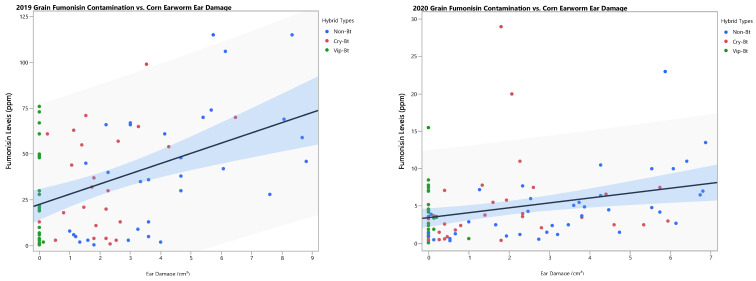
Linear regression analysis depicting the relationship between corn earworm ear damage and grain fumonisin contamination levels of field corn by year. 2019: R^2^ = 0.1970; *F* = 19.141; *p* < 0.0001. 2020: R^2^ = 0.0909; *F* = 9.3961; *p* = 0.0028.

**Table 1 insects-15-00914-t001:** Characteristics of all Bt and non-Bt field corn hybrids used in the 2019 and 2020 plantings.

Brand and Hybrid	Product Name	Bt Toxins ^a^	Year
DeKalb DKC 6694	Non-Bt	None	2019, 2020
DeKalb DKC 6697	Genuity VT Double PRO	Cry1A.105, Cry2Ab2	2019, 2020
DeKalb DKC 6629	Genuity Trecepta	Cry1A.105, Cry2Ab2, Vip3Aa20	2019, 2020
DeKalb DKC 6205	Non-Bt	None	2019, 2020
DeKalb DKC 6208	SmartStax	Cry1A.105, Cry2Ab2, Cry1Fa2	2019, 2020
DeKalb DKC 6824	Non-Bt	None	2020
DeKalb DKC 6826	Genuity VT Double PRO	Cry1A.105, Cry2Ab2	2020
DeKalb DKC 6799	Genuity Trecepta	Cry1A.105, Cry2Ab2, Vip3Aa20	2020
Pioneer 1637R	None	None	2019, 2020
Pioneer 1637YHR	Optimum Intrasect	Cry1Ab, Cry1Fa2	2019, 2020
Pioneer 1637VYHR	Optimum Leptra	Cry1Ab, Cry1Fa2, Vip3Aa20	2019
Pioneer 1870R	None	None	2020
Pioneer 1870YHR	Optimum Intrasect	Cry1Ab, Cry1Fa2	2020
Pioneer 2088R	None	None	2019, 2020
Pioneer 2089VYHR	Optimum Leptra	Cry1Ab, Cry1Fa2, Vip3Aa20	2019, 2020

^a^ All Bt hybrids also expressed glyphosate herbicide tolerance.

**Table 2 insects-15-00914-t002:** Effect of Bt traits on LS means ± SEM of percentage corn earworm-infested ears and damaged area by ear region per ear at growth stage R6 in 2019.

Brand and Hybrid ^a^	Bt Traits	Infested Ears at R3 (%)	Damaged Ears at R6 (%)	Damage (cm^2^) by Ear Region		
				Ear Tip	Kernels	Total
DKC 6694	None (RR2)	55.0 ± 5.2 c	57.5 ± 5.3 c	0.92 ± 0.14 e	1.15 ± 0.28 cd	2.07 ± 0.38 d
DKC 6697	VT Double PRO ^+^	28.3 ± 4.7 e	35.8 ± 6.7 d	0.47 ± 0.10 f	0.48 ± 0.15 ef	0.96 ± 0.22 e
DKC 6629	Trecepta ^++^	0 f	0 e	0 g	0 f	0 f
DKC 6205	None (RR2)	65.8 ± 6.1 b	61.7 ± 5.6 c	1.39 ± 0.16 cd	1.32 ± 0.26 cd	2.72 ± 0.35 cd
DKC 6208	SmartStax ^+^	41.7 ± 6.0 d	50.8 ± 4.2 c	1.15 ± 0.16 de	0.93 ± 0.19 de	2.08 ± 0.18 d
Pio 1637R	None (RR2)	84.2 ± 5.2 a	90.8 ± 3.3 a	2.37 ± 0.19 b	2.75 ± 0.57 b	5.11 ± 0.66 b
Pio 1637YHR	Intrasect ^+^	80.8 ± 4.8 a	77.5 ± 7.0 b	1.59 ± 0.30 c	1.60 ± 0.33 c	3.19 ± 0.57 c
Pio 1637VYHR	Leptra ^++^	0 f	0 e	0 g	0 f	0 f
Pio 2088R	None (RR2)	87.5 ± 5.4 a	96.7 ± 2.2 a	3.25 ± 0.16 a	3.47 ± 0.51 a	6.72 ± 0.57 a
Pio 2089VYHR	Leptra ^++^	1.7 ± 1.7 f	0 e	0 g	0 f	0 f
*F* > (*P*) (df = 9, 54)		96.98 (<0.0001)	78.72 (<0.0001)	28.74 (<0.0001)	26.49 (<0.0001)	55.84 (<0.0001)

LS means ± SEM within columns followed by the same letter are not significantly different (pair-wise *t*-tests of LSM, α = 0.05). ^a^ DKC for Dekalb hybrids; P for Pioneer brand hybrids. ^+^ Bt hybrid expressing only pyramided *Cry* proteins. ^++^ Bt-hybrid expressing pyramided *Cry* proteins with Vip3Aa20.

**Table 3 insects-15-00914-t003:** Effect of Bt traits on LS means ± SEM of percentage corn earworm-infested ears and damaged area by ear region per ear at R6 growth stage in 2020.

Brand and Hybrid ^a^	Bt Traits	Infested Ears at R3 (%)	Damaged Ears at R6 (%)	Damage (cm^2^) by Ear Region		
				Ear Tip	Kernels	Total
DKC 6694	None (RR2)	42.5 ± 14.3 bc	38.3 ± 14.3 bc	0.67 ± 0.26 cd	0.74 ± 0.28 de	1.42 ± 0.54 de
DKC 6697	VT Double PRO ^+^	39.2 ± 13.6 bc	28.3 ± 10.5 c	0.37 ± 0.14 d	0.42 ± 0.17 e	0.78 ± 0.31 e
DKC 6629	Trecepta ^++^	2.5 ± 2.5 d	0.8 ± 0.8 d	0.02 ± 0.02 e	0.01 ± 0.01 f	0.02 ± 0.02 f
DKC 6205	None (RR2)	56.7 ± 10.8 ab	55.8 ± 15.8 ab	1.01 ± 0.32 bc	1.48 ± 0.49 cd	2.49 ± 0.81 cd
DKC 6208	SmartStax^+^	36.8 ± 13.4 c	35.0 ± 9.9 c	0.64 ± 0.20 cd	0.83 ± 0.28 de	1.47 ± 0.48 de
DKC 6824	None (RR2)	55.0 ± 10.6 abc	60.8 ± 10.6 a	1.32 ± 0.28 ab	2.32 ± 0.44 b	3.64 ± 0.72 ab
DKC 6826	VT Double PRO ^+^	37.5 ± 10.9 c	35.8 ± 12.3 c	0.65 ± 0.25 cd	1.02 ± 0.45 de	1.67 ± 0.69 de
DKC 6799	Trecepta ^++^	3.3 ± 3.3 d	0.8 ± 0.8 d	0.01 ± 0.01 e	0.02 ± 0.02 f	0.02 ± 0.02 f
Pio 2088R	None (RR2)	49.2 ± 11.2 abc	65.0 ± 8.1 a	1.25 ± 0.21 ab	3.20 ± 0.62 a	4.45 ± 0.81 a
Pio 2089VYHR	Leptra ^++^	0	4.2 ± 2.5 d	0.11 ± 0.07 e	0.05 ± 0.05 f	0.16 ± 0.12 f
Pio 1637/1870R	None (RR2)	63.3 ± 11.1 a	71.7 ± 9.2 a	1.48 ± 0.27 a	2.42 ± 0.45 b	3.90 ± 0.70 ab
Pio 1637/1870YHR	Intrasect ^+^	54.2 ± 10.9 abc	54.2 ± 10.4 ab	1.12 ± 0.22 ab	2.19 ± 0.50 bc	3.31 ± 0.70 bc
*F* > (*P*) (df = 11, 66)		11.00 (<0.0001)	14.93 (<0.0001)	13.72 (<0.0001)	16.61 (<0.0001)	16.69 (<0.0001)

LS means ± SEM within columns followed by the same letter are not significantly different (pair-wise *t*-tests of LSM, α = 0.05). ^a^ DKC for Dekalb hybrids; P for Pioneer brand hybrids. ^+^ Bt hybrid expressing only pyramided *Cry* proteins. ^++^ Bt-hybrid expressing pyramided *Cry* proteins with Vip3Aa20.

**Table 4 insects-15-00914-t004:** Effect of Bt traits on LS means ± SEM of corn grain yield, test weights, aflatoxin contamination, and fumonisin contamination in 2019.

Brand and Hybrid ^a^	Bt Traits	Grain Yield (kg/ha)	Test Weight (kg/hL)	Aflatoxin (ppb)	Fumonisin (ppm)
DKC 6694	None (RR2)	14,732 ± 273 abcd	69.91 ± 1.67 a	57.00 ± 50.89 bc	32.00 ± 10.44 cde
DKC 6697	VT Double PRO ^+^	13,954 ± 594 d	70.71 ± 1.62 a	39.81 ± 25.08 bc	31.75 ± 10.47 def
DKC 6629	Trecepta ^++^	15,313 ± 651 abc	69.69 ± 1.74 a	3.31 ± 1.26 c	14.88 ± 6.07 f
DKC 6205	None (RR2)	15,658 ± 805 ab	70.11 ± 1.59 a	15.19 ± 10.96 bc	17.44 ± 6.44 ef
DKC 6208	SmartStax ^+^	15,456 ± 779 ab	70.17 ± 1.57 a	29.19 ± 16.58 bc	17.25 ± 5.66 ef
Pio 1637R	None (RR2)	15,140 ± 404 abc	68.63 ± 2.70 ab	13.19 ± 8.89 bc	39.50 ± 9.75 bcd
Pio 1637YHR	Intrasect ^+^	15,302 ± 477 abc	68.84 ± 2.06 ab	184.63 ± 56.43 a	57.13 ± 8.21 ab
Pio 1637VYHR	Leptra ^++^	14,201 ± 593 cd	70.28 ± 1.77 a	16.56 ± 12.53 bc	25.44 ± 9.52 ef
Pio 2088R	None (RR2)	14,672 ± 682 bcd	63.31 ± 2.30 c	120.38 ± 75.79 b	75.13 ± 11.26 a
Pio 2089VYHR	Leptra ^++^	15,830 ± 632 a	66.68 ± 2.55 b	116.98 ± 73.09 b	43.62 ± 9.15 abc
*F* > (*P*) (df = 9, 54)		2.33 (0.0255)	7.10 (<0.0001)	3.15 (0.0036)	8.80 (<0.0001)

LS means ± SEM within columns followed by the same letter are not significantly different (pair-wise *t*-tests of LSM, α = 0.05). Statistical analyses based on log_10_(X + 1) values. ^a^ DKC for Dekalb hybrids; P for Pioneer brand hybrids. ^+^ Bt hybrid expressing only pyramided *Cry* proteins. ^++^ Bt-hybrid expressing pyramided *Cry* proteins with Vip3Aa20.

**Table 5 insects-15-00914-t005:** Effect of Bt traits on LS means ± SEM of corn grain yield, test weights, aflatoxin contamination, and fumonisin contamination in 2020.

Brand and Hybrid ^a^	Bt Traits	Grain Yield (kg/ha)	Test Weight (kg/hL)	Aflatoxin (ppb)	Fumonisin (ppm)
DKC 6694	None (RR2)	12,029 ± 412 cd	75.04 ± 4.00 a	15.10 ± 14.99 bc	3.39 ± 0.87
DKC 6697	VT Double PRO ^+^	13,770 ± 686 a	70.29 ± 0.40 abc	8.73 ± 7.77 bc	4.15 ± 0.91
DKC 6629	Trecepta ^++^	13,478 ± 678 ab	70.00 ± 0.79 abc	0.01 ± 0.01 c	4.36 ± 0.76
DKC 6205	None (RR2)	12,498 ± 851 bcd	70.87 ± 1.78 abc	1.91 ± 1.59 bc	6.40 ± 2.65
DKC 6208	SmartStax ^+^	13,272 ± 932 ab	72.45 ± 4.12 ab	25.46 ± 18.91 bc	6.99 ± 3.40
DKC 6824	None (RR2)	13,049 ± 751 abc	68.81 ± 1.05 bcd	1.37 ± 0.94 bc	3.14 ± 1.15
DKC 6826	VT Double PRO ^+^	12,853 ± 652 abcd	68.91 ± 0.44 bcd	12.60 ± 12.49 bc	5.19 ± 2.18
DKC 6799	Trecepta ^++^	13,706 ± 797 a	69.54 ± 0.89 bcd	0.91 ± 0.62 bc	3.35 ± 0.93
Pio 2088R	None (RR2)	13,739 ± 556 a	64.59 ± 0.82 d	85.01 ± 59.23 a	6.44 ± 1.44
Pio 2089VYHR	Leptra ^++^	13,755 ± 727 a	66.24 ± 0.64 cd	58.26 ± 37.91 ab	5.63 ± 1.64
Pio 1637/1870R	None (RR2)	11,897 ± 1220 d	68.78 ± 1.47 bcd	0.29 ± 0.16 bc	4.27 ± 1.11
Pio 1637/1870YHR	Intrasect^+^	12,822 ± 376 abcd	70.19 ± 1.00 abc	53.14 ± 34.70 ab	3.50 ± 0.84
*F* > (*P*) (df = 11, 66)		3.37 (0.0008)	2.13 (0.0263)	2.35 (0.0150)	1.08 (0.3906)

LS means ± SEM within columns followed by the same letter are not significantly different (pair-wise *t*-tests of LSM, α = 0.05). Statistical analyses based on log_10_(X + 1) values. ^a^ DKC for Dekalb hybrids; P for Pioneer brand hybrids. ^+^ Bt hybrid expressing only pyramided *Cry* proteins. ^++^ Bt hybrid expressing pyramided *Cry* proteins with Vip3Aa20.

## Data Availability

The available data is presented in the paper and Appendix A.

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
