# Peer review of "Bt Trait Efficacy Against Corn Earworm, Helicoverpa zea, (Lepidoptera: Noctuidae) for Preserving Grain Yield and Reducing Mycotoxin Contamination of Field Corn"

_insects, 2024, doi:10.3390/insects15120914_

Round 1

Reviewer 1 Report

Comments and Suggestions for Authors

BT transgenic technology can prevent the damage of corn earworm in corn. This study evaluated the corn ear damage of different BT transgenic corn varieties to the ears of corn earworm and found that only Vip3Aa20 protein was effective and could effectively control the grain mycotoxin. And other toxins have no significant effect on corn ear damage. This provides more evidence for the use and shortcomings of BT genetically modified corn. The research is very interesting and meaningful, it is recommended to revise and accept it. There are suggestion for the study:

1. In the introduction, less emphasis was placed on the effects of Vip3Aa20 on Helicoverpa zea (Boddie), it’s better to added more introduce for the VIp3Aa20.

2.The introduction spends a lot of space discussing the effects of high temperatures and drought on corn kernels, but the subsequent experimental content section does not reflect this aspect. Consider whether these is needed.

3.The format of secondary titles is inconsistent. And some paragraph less space before the paraph.

4.Table 4 Table 5 must be in the same format. For example, the first two columns of Table 4 and Table 5 are aligned to the left and the middle three columns are aligned to the center, but the last table 4 is aligned to the center and Table 5 is aligned to the left. And the first column width needs to be adjusted, table header, data as far as possible in the same row.

5. The format of references is not uniform, and some references cite web addresses. Most of the literatures do not introduce DOI numbers, and some reference page numbers need to be supplemented.

6. Each paragraph of the text has different alignment methods, some choose to align both ends, some choose to align left, and the format must be unified.

Author Response

Reviewer #1 Comments

BT transgenic technology can prevent the damage of corn earworm in corn. This study evaluated the corn ear damage of different BT transgenic corn varieties to the ears of corn earworm and found that only Vip3Aa20 protein was effective and could effectively control the grain mycotoxin. And other toxins have no significant effect on corn ear damage. This provides more evidence for the use and shortcomings of BT genetically modified corn. The research is very interesting and meaningful, it is recommended to revise and accept it. There are suggestion for the study:

  1. In the introduction, less emphasis was placed on the effects of Vip3Aa20 on Helicoverpa zea(Boddie), it’s better to added more introduce for the VIp3Aa20.

 The Vip 3aA20 toxin is presented on lines 59-66. We added a sentence on lines 61-663 stating ‘Indeed, currently corn expressing the vip3Aa20 protein almost completely eliminates ear infestation by corn earworm’ We are not clear what additional information is being requested.

2.The introduction spends a lot of space discussing the effects of high temperatures and drought on corn kernels, but the subsequent experimental content section does not reflect this aspect. Consider whether these is needed.

 The sentence referring to temperature and drought stress was deleted at lines 85-87.  We do not see where else this topic is presented in the introduction.

3.The format of secondary titles is inconsistent. And some paragraph less space before the paragraph.

Secondary titles are corrected and spacing adjusted.  

 4.Table 4 Table 5 must be in the same format. For example, the first two columns of Table 4 and Table 5 are aligned to the left and the middle three columns are aligned to the center, but the last table 4 is aligned to the center and Table 5 is aligned to the left. And the first column width needs to be adjusted, table header, data as far as possible in the same row.

Columns in table are aligned to center the number around the plus/minus signs.

  1. The format of references is not uniform, and some references cite web addresses. Most of the literatures do not introduce DOI numbers, and some reference page numbers need to be supplemented.

References are updated and DOI numbers are added to all references where the numbers are available.

  1. Each paragraph of the text has different alignment methods, some choose to align both ends, some choose to align left, and the format must be unified.

The text is justified fully at both ends.

Reviewer 2 Report

Comments and Suggestions for Authors

Comments.

The article has good quality for publication; however, the authors center their results and Comments to article.

Title    Include.  Order: Family of the pest (Lepidoptera: Noctuidae)

Line 164-65. The central 2 rows for data collection were complete taken? or the row heads were not taken for data collection??

Table 4. Pio Leptra Aflatoxin contamination is high as compared with other results (119.98 ppb) is that correct? Even the same hybrid in 2020.

Line 383. Start a new paragraph and delete the word HERE.

Other journals require that every spp mentioned in the article should include (Order : Family) of the specie when first mentioned, Is this the case in Insecs???

Other discussion mainly on aflatoxins and mycotoxins rather than the effect of hybrids over the pest, ie, gallery length, number of larvae etc, data is available. ( Is this article more suitable to be published in a plant pathology journal???)

In the pathogens effects I suggest that severity and incidence indexes be included again, the data is available.

Author Response

Reviewer #2 Comments.

The article has good quality for publication; however, the authors center their results and Comments to article.

 Title    Include.  Order: Family of the pest (Lepidoptera: Noctuidae). Added to the title.

Line 164-65. The central 2 rows for data collection were complete taken? or the row heads were not taken for data collection??  

The plants were counted and rated visually for whorl damage by fall armyworm. The plants were not collected or damaged by the sampling. On line 170 it states that selected plants were inspected to verify identification of whorl-infesting larvae. We added the phrase ‘in border rows’ to clarify this point.  Plants within the plots were not damaged.

Table 4. Pio Leptra Aflatoxin contamination is high as compared with other results (119.98 ppb) is that correct? Even the same hybrid in 2020.

It is 116.98.  Yes numbers reported are correct in both years.  But the Pioneer 2089VYHR – Leptra did not have the largest level in the experiment. The Pioneer brand cultivars in general had greater aflatoxin levels than most of the DKC cultivars.  This is presumably related to the base genetics and ear characteristics of the two groups of cultivars and not to specific Bt traits. But there was a lot of variability in aflatoxin measurements as indicated by the relatively large SEM values.

Line 383. Start a new paragraph and delete the word HERE. 

Done

Other journals require that every spp mentioned in the article should include (Order : Family) of the specie when first mentioned, Is this the case in Insects???

The order and family names were added at first mention of an insect species.

Other discussion mainly on aflatoxins and mycotoxins rather than the effect of hybrids over the pest, ie, gallery length, number of larvae etc, data is available. ( Is this article more suitable to be published in a plant pathology journal???).

The paper presented a considerable amount of data on the insect response to Bt corn and the effect on yield.  It also includes the mycotoxin information.  The results of the insect efficacy and yield are not completely new, but the association between corn earworm ear injury and fumonisin contamination is not well documented so is a focus in the discussion. We feel the journal Insects is an appropriate outlet for this publication. Placing it in a plant pathology journal most likely would not be seen by many entomologists who are an important target audience for the work.

In the pathogens effects I suggest that severity and incidence indexes be included again, the data is available.

We did not measure the incidence and severity of infection before harvest.  A sample of the grain was processed as described in the paper to measure aflatoxin and fumonisin contamination of the grain and those data are presented.